# SE(3)-Equivariant Point Cloud-Based Place Recognition

**Chien Erh Lin, Jingwei Song, Ray Zhang, Minghan Zhu, and Maani Ghaffari**

University of Michigan, Ann Arbor, MI, USA

`{chienerh, jingweso, rzh, minghanz, maanigj}@umich.edu`

**Abstract:** This paper reports on a new 3D point cloud-based place recognition framework that uses SE(3)-equivariant networks to learn SE(3)-invariant global descriptors. We discover that, unlike existing methods, learned SE(3)-invariant global descriptors are more robust to matching inaccuracy and failure in severe rotation and translation configurations. Mobile robots undergo arbitrary rotational and translational movements. The SE(3)-invariant property ensures that the learned descriptors are robust to the rotation and translation changes in the robot pose and can represent the intrinsic geometric information of the scene. Furthermore, we have discovered that the attention module aids in the enhancement of performance while allowing significant downsampling. We evaluate the performance of the proposed framework on real-world data sets. The experimental results show that the proposed framework outperforms state-of-the-art baselines in various metrics, leading to a reliable point cloud-based place recognition network. We have open-sourced our code at https://github.com/UMich-CURLY/se3_equivariant_place_recognition.

**Keywords:** Place Recognition, SE(3)-Invariant, SE(3)-Equivariant Representation Learning, 3D Point Clouds

## 1  Introduction

Place recognition, also known as loop closure detection, can be defined as linking the sensor's in-situ observations and the prebuilt reference map. It is a critical component in Simultaneous Localization and Mapping (SLAM) and enables a robot to determine if it has seen a place before and provides loop closure candidates [1]. Among numerous robot perception sensors, 3D (stereo, LiDAR, and RGB-D) sensors are gaining popularity. They are widely equipped in modern service robots, autonomous cars [2], and drones [3] due to their better environment perception ability and decreasing prices. Thus, place recognition techniques with 3D data can be used in estimating the agent's location in scenarios such as self-driving vehicles, autonomous indoor navigation, or scientific exploration.

Extracting consistent features from 3D data is an important research topic but remains underexplored and unsolved [4]. One key issue in present place recognition methods is that they do not consider transformation changes in data or expect robustness via simple data augmentation [5]. Thus, the extracted global descriptors and their place recognition performance are sensitive to transformation variations in the training and testing point cloud samples. Our research aims at designing a rotation and translation-invariant global descriptor for point clouds, called SE(3)-invariant feature, to solve the transformation-sensitivity problem.

The attention mechanism of transformers [6] enables networks to learn the correlation between input features and obtain the importance of each feature. Some place recognition frameworks like PCAN [7] use an attention mechanism on local features to re-weight each feature. However, they usually apply an attention mechanism to feature space to learn the importance of each feature. In our work, we apply the attention mechanism on the 3D points to learn which point to reserve during the downsampling process.

In this paper, we propose a place recognition framework that exploits SE(3)-equivariant representation learning to perform place recognition in challenging rotation and translation scenes (Figure 1).

6th Conference on Robot Learning (CoRL 2022), Auckland, New Zealand.

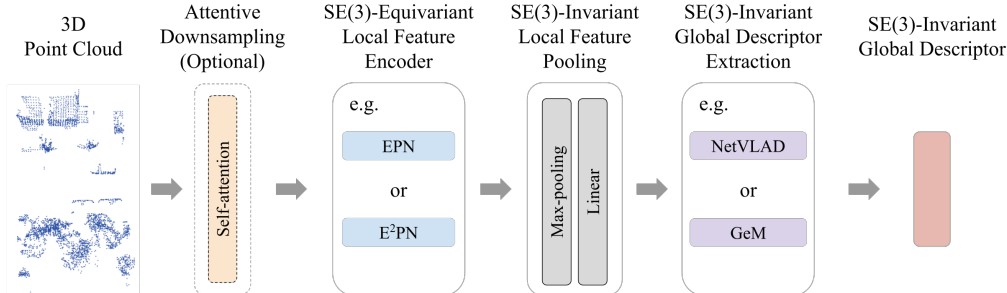

Figure 1: Overview of the proposed SE(3)-equivariant point cloud-based place recognition pipeline. Optionally, 3D point clouds are preprocessed with attentive downsampling. Next, SE(3)-equivariant local features are learned using SE(3)-equivariant networks. In this work, we leverage EPN and E$^2$PN. Then, SE(3)-invariant local features are extracted by max-pooling. Lastly, SE(3)-invariant global descriptors are computed by global pooling methods. In this work, we use NetVLAD and Generalized Mean (GeM). The global descriptors are SE(3)-invariant and can perform place recognition tasks.

We learn SE(3)-equivariant local features via group-equivariant encoder networks. And our pipeline is agnostic to the particular approach for learning equivariant features. Then, SE(3)-invariant global descriptors are learned by aggregating local features. Moreover, we apply a self-attention mechanism for downsampling point clouds to decrease memory usage and increase efficiency in training. We train and test our network on real-world data sets. We also validate our proposed framework for rotation and translation scenes. Experimental results show that our approach consistently outperforms existing state-of-the-art approaches. The experiment of the trained network on unseen data sets verifies the generalizability and scalability of our proposed framework.

The main contributions of this work can be summarized as follows:

1. We propose a new pipeline for place recognition using SE(3)-equivariant encoders to learn SE(3)-invariant descriptors with only geometric information from 3D point clouds. The proposed method is robust against arbitrary rotation and translation of robot poses. It is generalizable and scalable to unseen data sets, thereby removing the need for data augmentation.

2. We apply a self-attention mechanism to downsample point clouds which maintains place recognition in high performance up to 50% downsampling rate.

3. The code is open-sourced at `https://github.com/UMich-CURLY/se3_equivariant_place_recognition`.

## 2   Related Work

We first review place recognition works that use geometric information from 3D point clouds. Then, we present existing works that use point cloud descriptors with rotation-invariant or translation-invariant properties. Later, we discuss existing group-equivariant networks for 3D point clouds. At last, we provide a brief introduction to designing attention mechanisms to improve place recognition performance.

**Geometry-Based Place Recognition** Previously, place recognition using 3D point clouds relied on histograms or hand-engineered features such as Fast Histogram [8], M2DP [9] and Scan Context [10]. PointNetVLAD [11] is a pioneering work to apply a learning-based feature extractor to place recognition tasks. It combines local feature encoder PointNet [12] and global descriptor aggregator NetVLAD [13] to allow end-to-end representation training from a given 3D point cloud. Later, MinkLoc3D [14] presents sparse voxelized point cloud representation and sparse 3D convolutions. Other algorithms such as LPD-Net [15], OverlapNet [16], and LCDNet [17] provide different learning-based ways of constructing global descriptors for place recognition tasks. However, these algorithms are not robust to rotational and translational pose changes.

**Exploiting Symmetry in Place Recognition.** Considering that the observer may be in different orientations or locations, researchers propose some rotation-invariant hand-crafted descriptors such as histograms of range in a LiDAR scan [18], Scan Context [10, 19], frequency-domain Scan Context [20], LiDAR-Iris [21], and height coding descriptor [22] to perform place recognition. While these hand-crafted features are rotation-invariant, some structural information is ignored when composing them. Later, deep learning features are widely used since their performances surpass hand-crafted features [23].

Only a few works try to encode rotation-invariant or translation-invariant features into learning-based place recognition algorithms. PointNetVLAD [11] and LCDNet [17] try to increase robustness by randomly rotating input point clouds during training. To achieve yaw-angle-invariant, OverlapNet [16] and OverlapTransformer [24] use range images and Lu et al. [25] propose a RING descriptor. To achieve rotation-invariant, RINet [5] exploits additional semantic information with rotation equivariant convolution, SeqSphereVLAD [26] uses spherical convolution in spherical view, and RPR-Net [27] exploits rotation-invariant convolution. Nevertheless, these strategies do not consider both 3D rotation and translation differences in the pose, thus might not be sufficient in more challenging scenarios.

**Group-Equivariant Networks for 3D Point Clouds.** While a few works take rotation-equivariant and translation-equivariant properties into consideration in place recognition tasks, a series of works design network architectures with the equivariance property for general feature learning. For example, Spherical CNNs [28] and Vector Neuron [29] enable SO(3)-equivariant representation learning. Equivariant Point Network (EPN) [30] performs SE(3) separable convolution, which separates 6D convolution into convolutions in the 3D Euclidean space and in SO(3). $E^2PN$ [31] proposes a lightweight variant of SE(3)-equivariant network for point clouds. These networks generally address the group-equivariant feature learning problem in classification and segmentation tasks. They are only tested with point clouds in single object shapes but have not been tested extensively on 3D point clouds in real-world outdoor scenes. We propose a new pipeline for place recognition that exploits symmetry via group-equivariant networks. Since EPN and $E^2PN$ can encode SE(3)-equivariant information of point clouds, we adopt these two networks. The proposed framework is the first attempt to develop SE(3)-equivariant place recognition framework to bridge the gap between the group-equivariant and place recognition literature.

**Attention Mechanism in Place Recognition.** An attention mechanism has been applied to some place recognition tasks to learn the contextual features. PCAN [7], SOE-Net [32], Retriever [33], and OverlapTransformer [24] include this technique to learn the relations of different features. Among these applications, attention mechanisms are used to learn the importance of local features. In this work, we explore applying the attention mechanism to the input 3D points to learn the points' significance for downsampling.

## 3 Methodology

This section details our proposed framework for SE(3)-invariant place recognition using 3D point clouds. Figure 1 presents an overview of the proposed approach. The framework consists of three parts: attentive downsampling, SE(3)-invariant local feature extraction, and SE(3)-invariant global descriptor generation. We will fully discuss each component in the following subsections.

### 3.1 Attentive Downsampling

3D point clouds measured from LiDAR or RGB-D sensors may contain hundreds of thousands of points. To perform place recognition efficiently in neural networks, we exploit the attention mechanism to downsample point cloud measurements while preserving meaningful information. For a point cloud with $N$ points $P \in \mathbb{R}^{N \times 3}$, we apply the multi-head attention module [6] using the PyTorch library [34] to learn the attention weights $W_{atten} \in \mathbb{R}^{N \times 3}$ from the input point cloud. Multi-head attention is defined as $\text{MultiHead}(X) = \text{Concat}(head_1, head_2, head_3)W^O$, where $\text{Concat}(\cdot)$ does the features concatenation. The number of parallel attention heads is set as 3. Each attention head is defined as $head_i = \text{Attention}(XW_i^Q, XW_i^K, XW_i^V)$. Here, we set query $Q = XW_i^Q$, key $K = XW_i^K$, and value $V = XW_i^V$, where $X$ is the input point cloud $P$ to perform self-attention and learn the correlation between the input 3D points.

With the attention weights, we summarize over the feature space to obtain point-wise attention weights $W_{pw-atten} \in \mathbb{R}^N$, which represent the significance of each point. $W_{pw-atten} = \sum_{i=1,2,3} W_{atten}^i$, where $W_{atten}^i \in \mathbb{R}^N$ is the attention weight in dimension $i$. We select top-k attention weights and keep the corresponding points $P' = \mathbb{R}^{k \times 3}$.

## 3.2 Local SE(3)-Equivariant Features

Learning equivariant representation from point clouds can provide efficiency and generalizability in challenging robot perception tasks. *Equivariance* is a form of symmetry for functions that preserve the transformation applied on the input to the output.

Equivariance generalizes the concept of *invariance*, which means that the output of functions is independent of the transformations applied to the input. Mathematically, a function $f_{inv} : X \to Y$ is *invariant* to a set of transformations $T$, if for any $t \in T$, $f_{inv}(x) = f_{inv}(t \cdot x), \forall x \in X$.

The general linear group of degree $n$, denoted by $\mathrm{GL}_n(\mathbb{R})$, is the set of all $n \times n$ nonsingular real matrices, where the group binary operation is the ordinary matrix multiplication. The three-dimensional (3D) special orthogonal group, denoted by $\mathrm{SO}(3) = \{R \in \mathrm{GL}_3(\mathbb{R}) \mid RR^\mathsf{T} = I_3, \det(R) = +1\}$ is the rotation group on $\mathbb{R}^3$, where $I_3$ denotes the $3 \times 3$ identity matrix. The 3D special Euclidean group, denoted by $\mathrm{SE}(3) = \{H = (R, t) \mid R \in \mathrm{SO}(3), t \in \mathbb{R}^3\}$ is the group of rigid transformations, i.e., direct isometries on $\mathbb{R}^3$ [35].

In this work, we leverage Equivariant Point Network (EPN) [30] and E$^2$PN [31] to learn the SE(3)-equivariant feature and capture the inherent symmetry of 3D point cloud data. In the original EPN [30], given a 3D point $x$, a rotation $g$, a feature representation function $\mathcal{F} : \mathbb{R}^3 \times \mathrm{SO}(3) \to \mathbb{R}^D$, and a kernel $h : \mathbb{R}^3 \times \mathrm{SO}(3) \to \mathbb{R}^D$, the discretized SE(3)-equivariant convolutional operator is defined as the dot product between the translated and rotated kernel and the function $\mathcal{F}$: $(\mathcal{F} * h)(x, g) = \sum_{x_i \in \mathcal{P}} \sum_{g_j \in G} \mathcal{F}(x_i, g_j) h(g^{-1}(x - x_i), g_j^{-1} g)$, where $P$ and $G$ are the discretized sets corresponding to $\mathbb{R}^3$ and $\mathrm{SO}(3)$, respectively. To reduce the computation cost in 6D convolution, the authors separate the kernel $h$ into two smaller kernels representing SE(3) point convolution and SE(3) group convolution, respectively. This design preserves SE(3)-equivariant features from the input point cloud while maintaining affordable computation.

We also experimented with E$^2$PN [31], a lightweight and more efficient variant of EPN [30]. E$^2$PN leverages quotient representations to embed SO(3)-equivariance in a spherical feature space, resulting in much fewer feature dimensions than EPN. Therefore, it drastically reduces memory consumption and runtime while preserving the rotational equivariance. Such property is highly relevant to our task since we work with large-scale point clouds in outdoor environments.

## 3.3 Local SE(3)-Invariant Feature Pooling

After learning from an SE(3)-equivariant encoder, we obtain SE(3)-equivariant features $f_e(P) \in \mathbb{R}^{N \times C \times R}$, where $P$ is the input point cloud, $f_e(\cdot)$ is the mapping from point cloud to SE(3)-equivariant features, $N$ is the number of points, $C$ is the number of local features, and $R$ is the number of rotation group discretization. We then apply the pooling method to extract SE(3)-invariant features. To avoid the group attentive pooling failing if the point cloud is circularly symmetric as discussed in [30], we propose to apply max-pooling on the rotational dimension for each spatial point to generate SE(3)-invariant features and increase the robustness for different shapes of point clouds. In the max-pooling step, we only keep the maximum feature from one of the $R$ discretized rotation group elements. After max-pooling, the SE(3)-invariant feature is then represent as $f_{inv}(P) \in \mathbb{R}^{N \times C}$. The last part of the local feature extractor is a linear layer to map the SE(3)-invariant features to the desired dimension. See Figure 1 for an illustration.

## 3.4 Global SE(3)-Invariant Place Representation

Global descriptors are computed by aggregating local features using NetVLAD or Generalized Mean (GeM) [36]. NetVLAD learns cluster centers of VLAD (Vector of Locally Aggregated Descriptors) in a CNN framework. The output descriptors $V$ are defined as $V(j, k) = \sum_{i=1}^{N} \frac{e^{w_k^\mathsf{T} x_i + b_k}}{\sum_{k'} e^{w_{k'}^\mathsf{T} x_i + b_{k'}}} (x_i(j) - c_k(j))$. This equation shows $j$-th dimensions of the $i$-th

descriptor, where $x$ is the local feature. $w_k$, $b_k$, and $c_k$ are trainable parameters to learn the center of cluster $k$.

GeM is a trainable pooling layer that generalizes max-pooling and mean-pooling. With local feature input $x$, the GeM pooling output is defined as $f_{GeM} = (\frac{1}{|x|} \sum_{x_i \in x} x_i^p)^{\frac{1}{p}}$, where $p$ is a pooling parameter that can be learned or set manually.

To learn discriminative global descriptors, we use lazy quadruplet loss [11] to train the network to perform place recognition tasks. For each iteration of training, there is an anchor point cloud $P_a$, a "positive" point cloud $P_p$ that is similar to $P_a$, and a "negative" point cloud $\{P_n\}$ that are dissimilar to $P_a$, and a random point cloud in the training set $P_{n^*}$. The lazy quadruplet loss is defined as $Loss(P_a, P_p, P_n, P_{n^*}) = \max_j([\alpha + \delta_p \delta_{n_j}]_+) + \max_k([\beta + \delta_p \delta_{n_k^*}]_+)$, where $\alpha$ and $\beta$ are constant values to provide margin. The lazy quadruplet loss can minimize the L2 distance between the anchor and the positive representation $\delta_p = d(f(P_a), f(P_p))$ while maximizing the distance between the anchor and the negative representation $\delta_{n_j} = d(f(P_a), f(P_{n_j})), P_{n_j} \in \{P_n\}$.

## 4    Experimental Results and Discussion

We construct SE(3)-invariant place recognition descriptors using the described method. In this section, we examine the performance of place recognition, the SE(3)-invariant properties, and the design of attentive downsampling.

**Model Training.** We train our networks on Oxford RobotCar [37] benchmark created by Uy and Lee [11]. Oxford benchmark contains 45 sequences of a vehicle taking measurements using SICK LMS-151 2D LiDAR in similar routes for different times, days, and seasons. Each point cloud is a submap of a prebuilt map. The ground points are removed, and the point clouds are normalized to be zero mean and rescaled to within the range of [-1, 1]. Training and testing sets are geographically split with a ratio of 70 % and 30 %. For creating training tuples, a ground truth location within 10 meters is considered a positive sample, while a location more than 50 meters away is considered a negative sample. We train with 21,711 sub-maps. In EPN-NetVLAD, point clouds are first downsampled to 2048 points using the attention mechanism. For SE(3)-equivariant encoders (EPN and E²PN), we construct them with two layers of equivariant layers, one with 32 local features and one with 64 local features. The number of dimensions for the discrete rotation group is 60 in EPN and 12 in E²PN. The encoder is followed with max-pooling and a linear layer to map local features to 1024 dimensions. Then, we use NetVLAD or GeM to learn global descriptors with dimensions of 256. For GeM, we follow MinkLoc3D's structure and set the pooling parameter $p = 3$. The network is trained for 30 epochs with a learning rate of $5 \times 10^{-5}$. The hyper-parameters in lazy quadruplet loss are set as $\alpha = 0.5, \beta = 0.2$. The network parameters are optimized by ADAM [38].

Note that we do not need random rotation during the training process since the network is designed to generate the same descriptor as we rotate or translate the point cloud. The decreased need for data augmentation is an advantage of the proposed framework.

**Place Recognition Evaluation.** In place recognition tasks, precision and recall are the two well-established evaluation metrics [39]. Precision $= TP/(TP + FP)$ is the percentage of true loop closures among all the places we recognize. Recall $= TP/(TP + FN)$ is the percentage of places we recognize among all true loop closures. $TP$ is the number of true-positive cases, $FP$ represents the number of false-positive cases, and $FN$ stands for the number of false-positive cases. The F1 score is defined as F1 $= 2(\text{precision} \cdot \text{recall})(\text{precision} + \text{recall})$ to obtain a balancing metric between precision and recall.

**Place Recognition on Oxford benchmark.** We first evaluate the performance of the proposed method on the Oxford benchmark. The Oxford RobotCar data set consists of data collected by vehicles driving in a similar route at different times and seasons. Hence, every sequence revisits the path traveled by other sequences. When performing the evaluation, we generate the SE(3)-invariant global descriptor for each input point cloud. Then, we find the top 1, top 25, and top 1% of candidates' matches similar to the query point cloud in each sequence. We calculate the precision and recall rate, then average among different query point clouds in different sequences. The average recall curve represents the model performance for the top 25 matches. With these evaluation metrics and scikit-learn library [40], we report precision-recall curves, F1-recall curves, and average recall curves of the proposed method and other state-of-the-art methods are shown

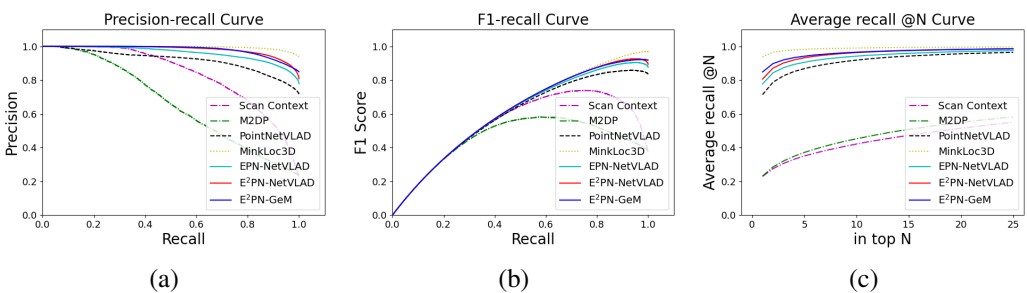

(a)              (b)              (c)

Figure 2: Place recognition evaluation on Oxford benchmark of the proposed methods and state-of-the-art approaches MinkLoc3D [14], PointNetVLAD [11], M2DP [9], and Scan Context [10].

Table 1: Experimental result showing the average recall (%) at top 1% and at top 1 on Oxford and in-house benchmark. Scan Context and M2DP are non-learning methods. Three methods in the middle rows use NetVLAD as a global pooling method. The last two methods in the bottom rows use GeM as a global pooling method. * is with attentive downsampling.

| | | Oxford | | U.S. | | R.A. | | B.D. | |
|---|---|---|---|---|---|---|---|---|---|
| | | AR@1% | AR@1 | AR@1% | AR@1 | AR@1% | AR@1 | AR@1% | AR@1 |
| hand-crafted | Scan Context [10] | 32.91 | 22.89 | 75.96 | 65.06 | 66.40 | 53.69 | 50.90 | 44.57 |
| | M2DP [9] | 34.69 | 23.14 | 45.03 | 32.41 | 44.62 | 34.34 | 39.34 | 32.95 |
| NetVLAD-based | PointNetVLAD [11] | 84.94 | 71.39 | 80.79 | 65.33 | 73.86 | 61.83 | 69.29 | 61.78 |
| | EPN-NetVLAD* | 89.17 | 77.69 | 87.82 | 74.03 | **81.98** | 70.09 | 76.91 | 69.14 |
| | $E^2$PN-NetVLAD | **91.83** | **82.46** | **88.04** | **84.73** | 80.58 | **80.93** | **79.84** | **79.31** |
| GeM-based | MinkLoc3D [14] | **97.91** | **93.76** | 95.04 | 86.01 | **91.19** | 81.17 | **88.48** | 82.66 |
| | $E^2$PN-GeM | 93.24 | 84.79 | **95.29** | **88.08** | 90.46 | **83.67** | 87.68 | **83.29** |

in Figure 2. All learning-based methods are trained on the same Oxford benchmark training set. However, MinkLoc3D is trained with a more efficient training strategy. Scan Context [10] and M2DP [9] construct hand-engineered features to perform place recognition. The result from Scan Context and M2DP can be further improved with hyper-parameter tuning. The figure shows that the proposed networks $E^2$PN-NetVLAD and EPN-NetVLAD outperform PointNetVLAD, all of which share the same global feature extraction method. MinkLoc3D and $E^2$PN-GeM both use GeM pooling for global feature extraction. Though MinkLoc3D performs the best among all methods, $E^2$PN-GeM and $E^2$PN-NetVLAD still perform consistently within 5% of difference.

**Place Recognition on in-house benchmark.** To show the generalizability of the proposed method, we also evaluate all methods on in-house data sets [11], which are constructed from Velodyne-64 LiDAR scans in three kinds of regions that are unseen to the network, including the university sector (U.S.), residential area (R.A.), and business district (B.D.). Table 1 shows the average recall at top 1% and at top 1 for each method on Oxford and in-house benchmark. Our method performs better than others for networks with NetVLAD global feature extraction regardless of selecting several or only one loop closure candidates. Our method achieves the best performance among all the data sets we did not train on. For methods that use GeM pooling, MinkLoc3D performs better on Oxford but similarly on U.S., R.A., and B.D. compared to the proposed $E^2$PN-GeM method, which shows the generalizability among data sets of the proposed method.

**KITTI benchmark.** We also evaluate the proposed methods on KITTI odometry data set [41]. Different from data in Oxford, 3D point clouds in the KITTI data set are collected by Velodyne HDL-64E, randomly downsampled to 4096 points, and the points of ground are not removed. The point clouds are scaled to [-1, 1] with zero mean. Sequence 00 is chosen for evaluation because of its large set of pairs for loop closure in the same orientation. While sequence 08 is chosen for evaluation because it contains 100% reverse loop closure. Reverse loop closures provide scenes with 180-degree viewing angle differences and advance a more challenging scenario. For sequence 00, the first 170 seconds construct the reference database, and the remaining part of the sequence is used as test queries. For sequence 08, the first 85 seconds and segment from 259 to 264 seconds construct the reference database, and the rest of the sequence is used as test queries. We ignore two nearby frames to avoid matching consecutive scans falsely. Table 2 reports the average recall at the top

Table 2: Average recall (%) at top 1% on KITTI sequence 00 and 08. * is with attentive downsampling.

| | KITTI Sequence 00 | | KITTI Sequence 08 | |
| | AR@1% | AR@1 | AR@1% | AR@1 |
|---|---|---|---|---|
| PointNetVLAD [11] | 73.18 | 70.68 | 32.47 | 17.61 |
| EPN-NetVLAD* (Ours) | 78.21 | 61.90 | 63.84 | 37.69 |
| E$^2$PN-NetVLAD (Ours) | **84.96** | **71.67** | **71.38** | **57.23** |
| MinkLoc3D [14] | 28.07 | 4.01 | 17.30 | 3.50 |
| E$^2$PN-GeM (Ours) | **86.22** | **77.94** | **71.70** | **59.75** |

Table 3: Experimental result showing the average recall (%) at top 1% of EPN-NetVLAD when the input point cloud is downsampled with different percentages and different methods on Oxford benchmark.

| Number of Points | 4096 | 3000 | 2048 | 1600 | 1024 |
|---|---|---|---|---|---|
| Downsampling Rate | 0 % | 27 % | 50 % | 61 % | 75 % |
| Random Downsampling | 71.66 | 63.34 | 57.29 | 53.19 | 43.17 |
| Attentive Downsampling | 71.66 | **71.65** | **71.05** | **66.22** | **57.97** |

Table 4: Data augmentation experiment on Oxford benchmark with different training data size and whether random transformation is applied during training. * is with attentive downsampling.

| | Random Transformation | Training Size: 3 Sequences | | Training Size: 45 Sequences | |
| | | AR@1% | AR@1 | AR@1% | AR@1 |
|---|---|---|---|---|---|
| PointNetVLAD [11] | | 69.38 | 54.00 | 86.88 | 73.12 |
| PointNetVLAD [11] | ✓ | 80.85 | 65.55 | 84.94 | 71.39 |
| EPN-NetVLAD * (Ours) | | 75.15 | 57.51 | 89.17 | 77.69 |
| E$^2$PN-NetVLAD (Ours) | | 85.16 | 70.61 | 93.78 | 85.04 |
| MinkLoc3D [14] | ✓ | - | - | 97.91 | 93.76 |
| E$^2$PN-GeM (Ours) | | 88.49 | 76.73 | 94.76 | 87.45 |

1% and top 1 for place recognition in sequence 00 and 08. All methods are trained using the same Oxford training data set. The table shows that the SE(3)-invariant property in the proposed methods helps them perform better in these challenging scenarios, supporting the better generalization claim.

**Attentive Downsampling.** We design an experiment to test the performance of point cloud downsampling using an attention mechanism. We study EPN-NetVLAD's performance with random and attentive downsampling methods. In this experiment, the network is constructed with only one layer of EPN with 64 local features and trained on three sequences of the Oxford data set to simplify the task. The test set includes point clouds in 23 sequences of the Oxford benchmark. The results of different downsamplings are presented in Table 3. It shows that an attention mechanism to downsample point clouds can maintain high place recognition performance up to 50% downsampling rate.

**Data Augmentation Experiment.** In Table 4, we experiment with different amounts of training data in the Oxford benchmark. PointNetVLAD relies on both random transformation and increasing training data size to achieve high performance, whereas E$^2$PN-NetVLAD can achieve similar performance with only training on three sequences. MinkLoc3D performs the best among all methods. However, it still requires random transformations in the training data.

**Descriptors with SE(3)-Invariant Property.** In addition to the place recognition experiments on Oxford and in-house benchmark, we take one point cloud in the Oxford benchmark, and construct simulated data by manually rotating and translating the point cloud to test the model performance with severe rotation and translation. We visualize the local features and global descriptors when the input point cloud is transformed under rotation, translation, rotated, then translated, and translated, then rotated. Figure 3 shows the results and the cosine similarity score between each transformed feature/descriptor and the original feature/descriptor. We can see that even if the point cloud is rotated or translated, the output features and descriptors remain the same and has 100 % similarity to the original one.

**Oxford Experiment with Random SE(3) Transformations.** We augment point clouds from the Oxford data set with random SE(3) transformations to create a more challenging place recognition experiment. The point clouds are transformed under purely SO(3)-rotation, purely 3D translation, and with both rotation and translation. With original point clouds in a range between [-1, 1], 3D rotations are applied randomly (uniformly sampled between $-\pi$ and $\pi$ along all three axes) using SciPy library [42], and 3D translations are applied with a standard deviation of 1.0. We use the model trained on 45 sequences of the Oxford benchmark to perform place recognition on this augmented data set which includes a total of 1320 point clouds that has 440 query point clouds, each of which has two positive pairs. Table 5 shows the average recall rate. We observe that with the attention downsampling module, the performance of EPN-NetVLAD decreases. The attentive downsampling module selects different points to feed into the network, whereas other methods compute over the same points in this case.

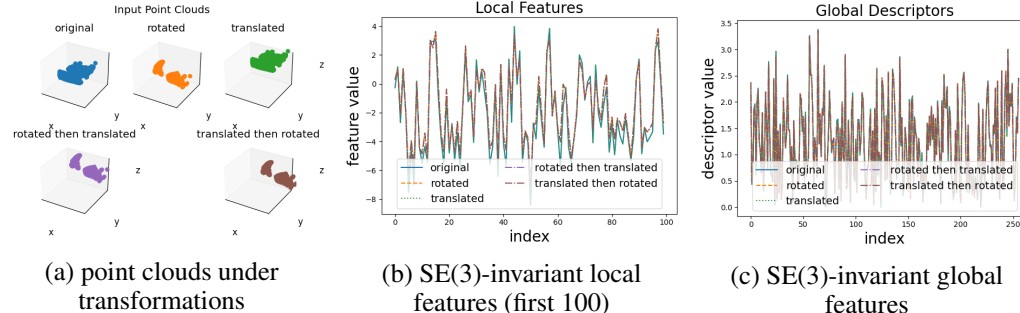

|   (a) point clouds under transformations   |   (b) SE(3)-invariant local features (first 100)   |   (c) SE(3)-invariant global features   |

Figure 3: Visualization of one input point cloud, the same point cloud under transformations, their corresponding local features, and the output global descriptors. There are 1024 local features for each point in the point cloud. We only show the first 100 features of the first point for this visualization.

Table 5: Average recall (%) at top 1% for performing place recognition task on simulated data where the point clouds are transformed under rotation or/and translation. * is with attentive downsampling.

Table 6: Run time performance per point cloud which contains 4096 points. The run times are computed without any network optimization. * attentive downsampling is performed to reduce point cloud size to 2048 points.

| Method | Rotation | Translation | Rotation and Translation |
|---|---|---|---|
| PointNetVLAD | 6.60 | 3.21 | 2.96 |
| MinkLoc3D | 13.71 | **100.00** | 13.77 |
| EPN-NetVLAD* | 30.13 | 40.69 | 27.55 |
| EPN-NetVLAD | 49.62 | **100.00** | 51.19 |
| $E^2$PN-NetVLAD | 71.38 | **100.00** | 68.81 |
| $E^2$PN-GeM | **75.97** | **100.00** | **73.91** |

| | Parameters | Run Time (s) |
|---|---|---|
| PointNetVLAD [11] | 19,779,145 | 0.006 |
| MinkLoc3D [14] | 1,055,713 | 0.005 |
| EPN-NetVLAD (Ours)* | 17,135,376 | 2.052 |
| $E^2$PN-NetVLAD (Ours) | 17,167,488 | 0.079 |
| $E^2$PN-GeM (Ours) | 192,513 | 0.082 |

**Run Time Performance.** We tested our method on a system equipped with an Intel i9-10900K CPU and an Nvidia GeForce RTX 3090. For obtaining one global descriptor from one 3D point cloud with 4096 points, we record the number of parameters in the network and report the run time performance per point cloud in Table 6. $E^2$PN-GeM has the lowest number of parameters. MinkLoc3D has the shortest inference time. Changing the global descriptor extraction method from NetVLAD to GeM drastically decrease the number of parameters but does not affect the run time substantially. We conjecture that the higher run times of SE(3)-equivariant networks are caused by the lack of network optimization. EPN and $E^2$PN are coded with custom functions to perform group convolutions, while other networks have network structures optimized on GPU. Thus, it is possible that SE(3)-equivariant networks can be further optimized to improve run time.

## 5   Limitation

The major limitation of the proposed framework is the relatively slow run time and the need for optimized libraries to perform real-time place recognition. However, with the development of more powerful computing hardware, we expect this limitation to be largely resolved in the near future. In addition, the study of equivariant encoders under other Lie groups to enable invariance to, e.g., scale and deformation is an interesting future direction that we did not discuss in this paper.

## 6   Conclusion

We have designed a place recognition framework that exploits SE(3)-equivariant representation learning. In particular, SE(3)-invariant features learned from 3D point clouds improve robustness to large transformations and generalizability in place recognition tasks. In addition, we propose using an attention mechanism in place recognition to downsample the input point cloud while maintaining high performance. Our experimental results on real-world data sets show the proposed method performs well in various metrics. Future work includes a lightweight design of the equivariant encoder for real-time onboard applications and the extension of this framework to stereo cameras where image data can also be incorporated into the learned representation.

**Acknowledgments**

Toyota Research Institute provided funds to support this work. Funding for M. Ghaffari was in part provided by NSF Award No. 2118818.

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
