# OpenReview forum: "SE(3)-Equivariant Point Cloud-Based Place Recognition"
_robot-learning.org/CoRL/2022/Conference — CoRL 2022 Poster_

### Official Review · Reviewer_cMpG · 2022-07-19

**Originality:** Good
**Technical Quality:** Very Good
**Clarity Of Presentation:** Excellent
**Impact:** 3

**Recommendation:**

Weak Accept: I recommend accepting the paper, but will not argue for my recommendation if the majority of other reviewers have a different opinion.

**Summary:**

This work proposed a place recognition method with robust pose invariance. It uses Equivariant Point Network (EPN) to learn SE(3)-equivariant features, which are further processed by NetVLAD to describe the place. Additionally, attentive downsampling is used to select representative points for improved computational efficiency. The training is based on the RobotCar dataset, and the evaluation uses standard metrics (PR and F1) for place recognition.

**Issues:**

- In 3.2, what is the motivation for replacing the group attentive pooling in EPN with max-pooling? Is it because of the attentive downsampling used in the first step?
- During the training process, do the authors also randomly rotate the input point cloud during training as PointNetVLAD and LCDNet do? Otherwise, how is the viewing angle invariance achieved?
- For the results in Table 1, what does the evaluation setup look like? How challenging (e.g., summer-winter place recognition) is it?
- In 4.5, the authors mentioned that the run-time is reduced from 2.05 seconds to 0.53 seconds when reducing discrete rotation groups from 60 to 20, how does the accuracy drop in this case?

**Quality Of The Limitations Section:**

Limitations are addressed clearly

**Reviewer Expertise:**

3: The reviewer is fairly confident that the evaluation is correct

**Robotics Focus:**

Highly relevant to robotics but no hardware experiments

**Strengths And Weaknesses:**

**Strengths**
- The paper is well written and easy to follow, the literature review is thorough, the methodology is clear and sound, and the experimental results are trustable.

**Weaknesses**
- The novelty of the method is slightly weak in the reviewer's mind. It seems like the system is a combination of three existing methods with a couple of minor modifications. Nevertheless, the combined system achieves outstanding results.
- The highlight of the method is high pose invariance for place recognition; however, the evaluation is mainly based on the RobotCar dataset, which is a driving dataset without significant viewing angle variation. It will be more persuasive if the authors include more challenging datasets to verify the pose invariance of the proposed method.

**Summary Of Recommendation:**

It is a well-written paper. The proposed method is clear and sound, although it slightly lacks of novelty. The evaluation is trustable, but it will be more persuasive if they include datasets with significant viewing-angle change.

---

> ### Author Response · Authors · 2022-08-27
> **Response to Reviewer cMpG**
>
> Thanks for your comments, suggestions, and questions!
>
> Along with the response below, we uploaded a revised version of the paper based on the comments from the reviewers. The major changes are highlighted in blue. We appreciate your help in making the paper better.
>
> **Q1**: In 3.2, what is the motivation for replacing the group attentive pooling in EPN with max-pooling? Is it because of the attentive downsampling used in the first step?
>
> **A1**: In EPN’s paper, the authors discuss that the group attentive pooling would fail if the point cloud is circularly symmetric. To increase the robustness for different shapes of point clouds, we decided to use max-pooling instead.
>
> **Q2**: During the training process, do the authors also randomly rotate the input point cloud during training as PointNetVLAD and LCDNet do? Otherwise, how is the viewing angle invariance achieved?
>
> **A2**: We do not need random rotation during the training process since the network is designed to generate the same descriptor as we rotate or translate the point cloud. The decreased need for data augmentation is an advantage of the proposed framework. More discussion about data augmentation is included in the revised paper, section 4.2.3.
>
> **Q3**: For the results in Table 1, what does the evaluation setup look like? How challenging (e.g., summer-winter place recognition) is it?
>
> **A3**: The Oxford RoboCar Dataset consists of data collected by vehicles driving in a similar route at different times and seasons. Hence, every sequence revisits the path traveled by other sequences. During evaluation, we generate the SE(3)-invariant global descriptor for each input point cloud. Then, we find the top 1 and top 1% of candidates similar to the query point cloud in each sequence. We calculate the precision, recall rate, and average among different query point clouds in different sequences. The detailed explanation is revised in section 4.2.1. Since we are using 3D point clouds as measurements, the appearance changes between seasons are not obvious in the measurement.
>
> **Q4**: In 4.5, the authors mentioned that the run-time is reduced from 2.05 seconds to 0.53 seconds when reducing discrete rotation groups from 60 to 20, how does the accuracy drop in this case?
>
> **A4**: The table below shows the comparison of the proposed network using a different number of rotation groups. Reducing the number of rotation groups from 60 to 20 increases efficiency while maintaining the place recognition performance on Oxford. We also tested on KITTI sequence 08, where reverse loop closure exists and provides a more challenging scenario. The performance of using 20 rotation group drops but still outperforms PointNetVLAD. More results of the run time are added to Table 6 in the revised paper.
>
> |                  | Inference Time (s) | Average recall at top 1 % on Oxford | Average recall at top 1 % on KITTI sequence 08 |
> |------------------|--------------------|-------------------------------------|------------------------------------------------|
> | PointNetVLAD     | 0.006              | 84.94 %                             | 32.47%                                         |
> | EPN-NetVLAD (60) | 2.052              | 89.17 %                             | 63.84%                                         |
> | EPN-NetVLAD (20) | 0.534              | 90.60 %                             | 46.84%                                         |
>
>
> **Q5**: It will be more persuasive if the authors include more challenging datasets to verify the pose invariance of the proposed method.
>
> **A5**:  In addition to the Oxford RoboCar Dataset, we also experimented on the KITTI dataset. The following table reports the average recall at the top 1% for place recognition in sequence 00 and sequence 08. Both methods are trained using the same Oxford training dataset. In sequence 08, reverse loop closure exists where there are revisiting the same place with 180-degree viewing angle differences. The table shows that EPN-NetVLAD’s SE(3)-invariant property helps it perform better in these challenging scenarios. More results are added to section 4.2.2 in the revised paper.
>
> |                    | KITTI Sequence 00 (loop closure in the same direction) | KITTI Sequence 08 (loop closure in the reverse direction) |
> |--------------------|--------------------------------------------------------|-----------------------------------------------------------|
> | PointNetVLAD       | 73.18%                                                 | 32.47%                                                    |
> | EPN-NetVLAD (Ours) | 78.21%                                                 | 63.84%                                                    |

---

### Official Review · Reviewer_kN1U · 2022-07-26

**Originality:** Good
**Technical Quality:** Very Good
**Clarity Of Presentation:** Good
**Impact:** 4

**Recommendation:**

Strong Accept: I recommend accepting the paper and will argue for my recommendation even if other reviewers hold a different opinion.

**Summary:**

The paper presents an equivariant SE3 transformations based architecture (building on existing EPN model) and NetVLAD global pooling (existing work) with quadruplet margin based contrastive loss (existing work). Standard benchmark datasets are used to show improved results against existing methods.
The main contribution of the paper is the combination of the above mentioned components to achieve invariant global features. Secondly, the proposed attentive downsampling is shown to be more robust than random downsampling of the input data.

**Issues:**

As already summarized above, additional experimental analyses and improved description of the method (as pointed above) will be critical to improve the rating of this paper.

**Quality Of The Limitations Section:**

Limitations are addressed clearly

**Reviewer Expertise:**

5: The reviewer is absolutely certain that the evaluation is correct and very familiar with the relevant literature

**Robotics Focus:**

Relevant but unlikely to deploy to hardware in near future

**Strengths And Weaknesses:**

- The main strength of the paper is the demonstration that combining the existing components, that is, equivariant network + group pool, global pool and quadruplet loss, can lead to SE3 transformation robust global features.
- The attentive downsampling is shown to be superior to random downsampling.
- The paper is well written and easy to understand.

Weaknesses and areas of improvement are listed below:

**Claims**

The contribution claim 1 and 2 overlap in discussing SE3 invariance. Also, claim 2 needs to be justified for scalability (what does it mean here?). Moreover, isn’t generalizability a must for machine learning based methods, should this really be a contribution?

**Method**

Section 3.1 is somewhat misleading. If Q,K,V,P are all the same, then there remains no learnable weight because by definition attention A = Q.K, and P_out = A.V, where typically Q,K,V are all learnable linear transformations of P. The authors seem to have made a direct use of Pytorch library functions without reflecting their understanding of what is being done under the name of ‘attention’, which in this case could have been heavily simplified if there are no learnable parameters and if Q,K,V,P are all the same.

Section 3.2 explains an existing method with modifications listed in L172-174. The authors mention that they replace group pooling with “max-pooling in rotation dimension for each spatial point”. So, what exactly is different here as the original group pooling would also have been applied in the rotational dimension. Is switching to max-pool operation better than what was originally used?


**Results and Analyses**

*Benchmarking*:
L88 critiques existing works arguing about their lack of robustness against translation and rotation but this has not been extensively demonstrated through benchmarking.

The authors did not compare against MinkLoc3D and other recent methods (see https://github.com/jac99/MinkLoc3D). PointNetVLAD’s results have been outperformed by a significant margin now. Also, MLP in PointNetVLAD and 3D sparse convolution in MinkLoc3D carry different inductive biases in training, where the former is understood to be more likely to fail under 3D transformations.

*Rotation vs Translation*:
As mentioned in the previous comment, convolution based learning such as that in MinkLoc3D should be able to handle 3D translation. The authors should consider presenting results separately for translation and rotation on top of the combined results.

*Global Pooling*:
VLAD vs GeM is another interesting question as VLAD can be more resource consuming and is more complicated than GeM.

*Simple baseline*:
Since the ultimate goal of the work is to achieve invariance to SE3 transformations, the authors should compare against a naive baseline where an equivalent standard CNN (MinkLoc3D?) is used but with significant training data augmentation (similar to the test set referred to in Section 4.3). This will highlight whether or not achieving invariance via equivariance is necessary or not.

*Discretized Rotation Angles*:
Equivariant networks operate on a discrete domain, which means that performance may be low for the rotation angles that were not considered during training. The authors should show performance against rotation angles and mention such limitations if existing.

**Minor**

- L29, many 3d place recognition methods already exist, as mentioned in Section 2
- L35, what “transformation changes”
- L46, local features are defined on and represent 3D points and their neighborhoods, L46-47 need to provide a more clear argument
- SeqSphereVLAD [a] is a closely related work which the authors can consider reviewing.

[a] Yin, Peng, et al. "Seqspherevlad: Sequence matching enhanced orientation-invariant place recognition." 2020 IEEE/RSJ International Conference on Intelligent Robots and Systems (IROS). IEEE, 2020.


**Summary Of Recommendation:**

Benchmarking is not strong in this paper and it seems that some of the recent methods will likely perform much better than the baselines considered in this work. Furthermore, an important simple baseline with trainind data augmentation might render computationally-expensive equivariant networks useless. Attentive downsampling, being one of the novel contributions, is not described well. Combination of existing components to come up with the proposed method is good engineering advance but the paper lacks any insights into aspects like inductive biases of different architectures (MLP, 3D convolutions, equivariant networks), rotation vs translation, and importantly, comparison against different training strategy such as training data augmentation.

Post-rebuttal:
The authors have done an amazing job at addressing all the concerns so raised. This includes clarifications on claims and methodology and more importantly, a range of new experiments which has significantly improved the paper.

The overall results on different benchmark datasets seem to be mixed, some in favor of the proposed method and others against (compared to MinkLoc3D). However, Table 4 has perhaps the strongest results in the authors' favor, which puts the inherent (and desired) characteristics of the proposed method at the forefront, that is, robustness to rotations.

The authors could perhaps have also included rotation transformations on datasets other than the simulated one and KITTI to show an expected performance drop for MinkLoc3D, e.g. by simulating an opposite direction travel.

Even though the paper builds on existing works, it currently boasts a well executed set of experiments that highlight the limitations of existing works and datasets while proposing an alternative approach tuned to viewpoint/orientation equivariance/invariance characteristics, thus, an improved rating.

---

> ### Author Response · Authors · 2022-08-27
> **Response to Reviewer kN1U**
>
> Thanks for your comments, suggestions, and questions!
>
> Along with the response below, we uploaded a revised version of the paper based on the comments from the reviewers. We included more experiments using different SE(3)-equivariant networks with other feature extraction methods. The major changes are highlighted in blue. We appreciate your help in making the paper better.
>
> **Q1**: The contribution claim 1 and 2 overlap in discussing SE3 invariance.
>
> **A1**: Thank you for the comment. We agree that those two contributions overlap. Thus we merge them into one.
>
> **Q2**: What does scalability mean in the contribution claim 2? Isn’t generalizability a must for machine learning based methods, should this really be a contribution?
>
> **A2**: The scalability mentioned in the contribution means that we can apply this method without excessive data augmentation to obtain robustness in pose changes. Therefore, we can deploy the model in a broader range of applications. Generalizability is an ideal goal for most machine learning methods. Testing on different datasets that the network is not trained on shows the generalizability of the proposed method.
>
> **Q3**: Explanation on self-attention. If Q,K,V,P are all the same, then there remains no learnable weight because by definition attention A = Q.K, and P_out = A.V, where typically Q,K,V are all learnable linear transformations of P.
>
> **A3**: Sorry for the confusion. For self-attention, query Q, value V, and key K have the same input X=P which is multiplied by different learnable weights ($W^Q$, $W^V$, and $W^K$). Thus, query $Q = X W^Q$, key $K = X W^K$, and value $V = X W^V$. We made some adjustments to the notation in the revised paper, section 3.1.
>
> **Q4**: What is the difference between EPN’s group pooling and the max-pooling used in the paper? Is switching to max-pool operation better than what was originally used?
>
> **A4**: In EPN’s paper, the authors discuss that the group attentive pooling would fail if the point cloud is circularly symmetric. To increase the robustness for different shapes of point clouds, we decided to use max-pooling instead.
>
> **Q5**: The authors did not compare against MinkLoc3D and other recent methods.
>
> **A5**: Thanks for the suggestion. We added the MinkLoc3D result in the revised paper and compared it with the proposed framework in different metrics.
>
> **Q6**: The authors should consider presenting results separately for translation and rotation on top of the combined results.
>
> **A6**: Thanks for the suggestion. Our experiment in section 4.3 separately tests the network performance on severe rotation and translation.
>
> **Q7**: VLAD vs GeM is another interesting question as VLAD can be more resource consuming and is more complicated than GeM.
>
> **A7**: Thanks for the suggestion. We experimented with a model of using E$^2$PN and GeM to test this global pooling method and include the result in the revised paper.
>
> **Q8**: The authors should compare against a naive baseline where an equivalent standard CNN is used but with significant training data augmentation. This will highlight whether or not achieving invariance via equivariance is necessary or not.
>
> **A8**: Thank you for the suggestion. We added a data augmentation experiment with different training data sizes in section 4.2.3. We also compared whether or not the model requires random transformation during training.
>
> **Q9**: The authors should show performance against rotation angles and mention if there exists limitations on discretized rotational angles in EPN.
>
> **A9**: Thank you for the suggestion. In our experiment with SE(3) transformation in section 4.3, we generate data with random rotation. Table 4 shows that EPN is robust in most possible rotations.

---

### Official Review · Reviewer_GJ8U · 2022-07-29

**Originality:** Fair
**Technical Quality:** Good
**Clarity Of Presentation:** Fair
**Impact:** 3

**Recommendation:**

Weak Reject: I recommend rejecting the paper, but will not argue for my recommendation if the majority of other reviewers have a different opinion.

**Summary:**

This paper addresses the problem of location matching with LiDAR point clouds, and proposes EPN-NetVALD using SE(3)-invariant global features. It consists of three major components: 1. Self-attention based point cloud downsampler; 2. Equivariant Point Network for point cloud equivariant feature extraction; 3. NetVLAD, a global feature extractor.  EPN extractor uses two separate kernels to perform SE(3) point convolution and SE(3) group convolution and generates SE(3)-equivariant local features which are aggregated into SE(3)-invariant local features by pooling. NetVLAD then clusters the local features into a single global descriptor by finding the center of local features.  The paper demonstrates that the proposed method outperforms the current STOA method on Oxford RobotCar dataset.

**Issues:**

1. Since this paper is all about SE(3) equivariant and invariant, the related work and method section needs to go into more detail about the SE(3) Lie group.
2. For place recognization for an unchanged scene,  changing the viewing angle not only introduces rotation and translation but also changes the observed areas. Which is not discussed in this paper.
3. In the method section, the extraction of invariant features is unclear, what is the rotational dimension and what is the neighborhood the max pooling is acted on?

**Quality Of The Limitations Section:**

Additional details required

**Reviewer Expertise:**

4: The reviewer is confident but not absolutely certain that the evaluation is correct

**Robotics Focus:**

Highly relevant to robotics but no hardware experiments

**Strengths And Weaknesses:**

Strengths:
1.  The idea of introducing SE(3) equivariant learning into point cloud and scene recognization is novel and interesting. Given an unchanged scene, the changes introduced by different viewing angle can be considered as a rotation and translation transformation of the scene.  Which can be captured by a SE(3) invariant descriptor, and the invariance considerably reduces the task difficulty for the model as rotation and translation mapping don't need to be learned during training.
2. The ablation studies for self-attention downsampler vs random downsampler and rotation and translation augmented dataset for testing SE(3)-invariance are good, as they clearly demonstrate the contribution of downsampler and SE(3)-invariant descriptors towards overall model performance.

Weakness:
1. The way to extract SE(3) invariant features are questionable, since only the invariant features are used for global feature learning as well as place matching, the necessity of introducing SE(3) equivariant network is unclear, especially when EPN is the bottleneck of the overall network runtime. Instead, for example in SE(3) transformer, only the 0 type spherical harmonic needs to be pooled to extract the invariant features.
2. For place recognization for an unchanged scene,  changing the viewing angle not only introduces rotation and translation but also changes the observed areas. Which is not discussed in this paper.
3. In the method section, the extraction of invariant features is unclear, what is the rotational dimension and what is the neighborhood the max pooling is acted on?

**Summary Of Recommendation:**

This paper addresses the problem of location matching with LiDAR point clouds, and proposes EPN-NetVALD using SE(3)-invariant global features. The idea of introducing SE(3) equvariant learning into point cloud and scene recognization is novel and interesting, as the viewing angle changes can be considered as observing a rotated and translated scene. This paper also demonstrates EPN-NetVALD STOA performance and has a clear ablation study of contribution of model's separate components.

However, although SE(3) equivariant and invariant learning is the main idea for this paper, it doesn't go into the details of SE(3) theory and how each such Lie group can be used for equivariant and invariant feature extraction. Additionally, since EPN-NetVALD's local feature extractor is built on EPN with minor modifications (only the pooling layer), the technical details of this part is unclear. Given NetVALD is also well established by previous methods, combining two networks into one pipeline makes me concerned bout this paper's novelty.

---

> ### Author Response · Authors · 2022-08-27
> **Response to Reviewer GJ8U**
>
> Thank you for your comments, suggestions, and questions!
>
> Along with the response below, we uploaded a revised version of the paper based on the comments from the reviewers. We included more experiments using different SE(3)-equivariant networks with other feature extraction methods. The major changes are highlighted in blue. We appreciate your help in making the paper better.
>
> **Q1**: Need more explanation on SE(3) invariant features extraction. What is the rotational dimension and what is the neighborhood the max pooling is acted on?
>
> **A1**: After learning SE(3)-equivariant features, we obtain feature $f_e(P) \in \mathbb{R}^{N \times C \times R}$, where $P$ is the input point cloud, $f_e()$ is the mapping function from point cloud to equivariant features, $N$ is number of points, $C$ is number of local features, and $R$ is number of rotation groups. In the max-pooling step, we only keep the maximum feature from one of the 60 discretized rotation groups for max-pooling. After max-pooling, the SE(3)-invariant feature is then $f_{inv}(P) \in \mathbb{R}^{N \times C}$. We added this additional explanation to section 3.3 in the paper.
>
> **Q2**: Need some discussion in the change of observed areas.
>
> **A2**: Thanks for pointing it out. Yes, the observed area does change when the robot pose is changed. In all the real-world datasets we tested on (Oxford, in-house, and KITTI), all loop closure pairs have different observed areas, whereas they can still perform place recognition.
>
> **Q3**: Related work and method section needs to go into more detail about the SE(3) Lie group.
>
> **A3**: Thanks for the advice. We added a paragraph in section 3.2 to explain the SE(3) Lie group.

---

### Official Review · Reviewer_xfgJ · 2022-08-07

**Originality:** Good
**Technical Quality:** Good
**Clarity Of Presentation:** Very Good
**Impact:** 3

**Recommendation:**

Weak Reject: I recommend rejecting the paper, but will not argue for my recommendation if the majority of other reviewers have a different opinion.

**Summary:**

This paper presents EPN-NetVLAD, a method for 3D point cloud based place recognition using SE(3)-Invariant features.

The key design of EPN-NetVLAD is that it leverages the Equivariant Point Network (EPN) proposed in [13] to learn local SE(3)-equivariant features, and then uses a global pooling to obtain SE(3)-invariant features.

Experiments show that EPN-NetVLAD outperforms existing methods that do not consider SE(3)-invariance, especially when the point clouds undergo large rotation and translation.

**Issues:**

N/A

**Quality Of The Limitations Section:**

Limitations are addressed clearly

**Reviewer Expertise:**

3: The reviewer is fairly confident that the evaluation is correct

**Robotics Focus:**

Highly relevant to robotics but no hardware experiments

**Strengths And Weaknesses:**

Strengths:
- Leveraging SE(3)-equivariant features for more robust place recognition
- Paper is easy to read, with clear motivation and intuition

Weakness
- I feel there is not much technical contribution in this paper. It is basically plugging the SE(3)-equivariant network into place recognition (even though this paper is the first to do so, I dont think this is a surprising idea and result).

**Summary Of Recommendation:**

I personally think more technical meat is needed to meet my calibration of an accepted paper in CoRL.
However, I would not strongly object if this paper is accepted. Overall a good paper making some contribution to the field.

---

> ### Author Response · Authors · 2022-08-27
> **Response to Reviewer xfgJ**
>
> Thanks for your comments!
>
> Along with the response below, we uploaded a revised version of the paper based on the comments from the reviewers. The major changes are highlighted in blue. We appreciate your help in making the paper better.
>
> Q1: Need more technical contribution other than plugging the SE(3)-equivariant network into place recognition.
>
> A1: We revised our paper and included more experiments using different SE(3)-equivariant networks with other feature extraction methods. From the literature review, group-equivariant networks were mainly tested on classification and segmentation tasks but not place recognition. In addition, they are usually tested with point clouds of single object shapes but not point clouds in real-world outdoor scenes. We want to highlight that our work is the first work that addresses the gap between group-equivariant and place recognition. Although geometric deep learning has been a focused topic for more than five years, existing place recognition works have not embraced group-equivariant networks into the framework. Therefore, even though our proposed framework is not surprising, it is a logical step moving forward.

---

### Author Response · Authors · 2022-08-27
**Revised Paper - SE(3)-Equivariant Point Cloud-based Place Recognition**

We thank all reviewers for their comments and suggestions. We have revised our paper and included more experiments using different SE(3)-equivariant networks with other feature extraction methods. The major changes are highlighted in blue.

---

### Author Response · Authors · 2022-08-28
**New Revision**

We updated E$^2$PN-NetVLAD and E$^2$PN-GeM results in Table 4 for SE(3) transformation experiment. The constraint on the kernel is modified in E$^2$PN for E$^2$PN-NetVLAD and E$^2$PN-GeM to achieve better robustness in severe SE(3) transformation.

---

### Meta-Review · Area_Chair_ZpP1 · 2022-08-15

**Recommendation:** Accept (Poster)
**Confidence:** 4

**Metareview:**

Main strengths:
1. Introducing SE(3) equivariance into 3D descriptor learning is a nice contribution.

Main weaknesses:
1. Novelty needs to be further justified.
2. Core premise of the method needs further explanations, particularly on whether the method is practical in real-life applications that will experience different observable areas/occlusions when the viewing poses change.
3. Several technical details need to be further clarified.
4. Experiments/benchmarking are not fully convincing.

The clarifications on methodology and new results provided in the rebuttal have addressed the major concerns raised by the reviewers, thus an accepted outcome is justified.

---

> ### Author Response · Authors · 2022-08-27
> **Response to Area Chair ZpP1**
>
> We thank all reviewers’ constructive comments. Along with the response below, we uploaded a revised version of the paper based on the comments from the reviewers. We appreciate your help in making the paper better.
>
> 1. **Novelty needs to be further justified.** We uploaded a revised paper and included more experiments using different SE(3)-equivariant networks with other feature extraction methods. From the literature review, group-equivariant networks were mainly tested on classification and segmentation tasks but not place recognition. We want to highlight that our work is the first work that addresses the gap between group-equivariant and place recognition.
> 2. **Core premise of the method needs further explanations** More explanation of the method is added in the revised paper.
> 3. **Several technical details need to be further clarified.** More technical details are added in the revised paper.
> 4. **Experiments/benchmarking are not fully convincing.** We added additional experiments/benchmarks in the revised paper to show the proposed pipeline’s performance and generalizability. We tested on KITTI dataset, conducted data augmentation experiment, and compared with another learning-based method.